# Formation of Superhydrophobic Coatings Based on Dispersion Compositions of Hexyl Methacrylate Copolymers with Glycidyl Methacrylate and Silica Nanoparticles

**DOI:** 10.3390/polym16213094

**Published:** 2024-11-01

**Authors:** Viktor V. Klimov, Alexey K. Shilin, Daniil A. Kusakovskiy, Olga V. Kolyaganova, Valentin O. Kharlamov, Alexander V. Rudnev, Manh D. Le, Evgeny V. Bryuzgin, Alexander V. Navrotskii

**Affiliations:** 1Chemical Engineering Faculty, Volgograd State Technical University, 28 Lenin Ave, 400005 Volgograd, Russia; shilin_alex@mail.ru (A.K.S.); k4isern@yandex.ru (D.A.K.); ollik86@mail.ru (O.V.K.); harlamov_vo@mail.ru (V.O.K.); bryuzgin_e@mail.ru (E.V.B.); a-navrotskiy@yandex.ru (A.V.N.); 2A.N. Frumkin Institute of Physical Chemistry and Electrochemistry of the Russian Academy of Sciences, Leninsky Prospekt, 31, Building 4, 119071 Moscow, Russia; av.rudnev@gmail.com; 3Southern Branch of Joint Vietnam-Russia Tropical Science and Technology Research Center, 3, 3/2 Str., District 10, Ho Chi Minh City 740300, Vietnam; ducmanh89@gmail.com

**Keywords:** polymer coatings, methacrylic copolymers, superhydrophobicity, roughness, nanoparticles

## Abstract

In the last decade, the task of developing environmentally friendly and cost-effective methods for obtaining stable superhydrophobic coatings has become topical. In this study, we examined the effect of the concentrations of filler and polymer binder on the hydrophobic properties and surface roughness of composite coatings made from organic–aqueous compositions based on hexyl methacrylate (HMA) and glycidyl methacrylate (GMA) copolymers. Silicon dioxide nanoparticles were used as a filler. A single-stage “all-in-one” aerosol application method was used to form the coatings without additional intermediate steps for attaching the adhesive layer or texturing the substrate surface, as well as pre-modification of the surface of filler nanoparticles. As the ratio of the mass fraction of polymer binder (Wn) to filler (Wp) increases, the coatings show the lowest roll-off angles among the whole range of samples studied. Coatings with an optimal mass fraction ratio (Wn/Wp = 1.2 ÷ 1.6) of the filler to polymer binder maintained superhydrophobic properties for 24 h in contact with a drop of water in a chamber saturated with water vapor and exhibited roll-off angles of 6.1° ± 1°.

## 1. Introduction

Superhydrophilicity and superhydrophobicity represent the boundary physicochemical states of a surface in contact with water. Materials allowing effective control of surface wettability have high potential for industrial applications and have been attracting the attention of scientists for over two decades [1,2]. On the superhydrophilic surfaces, water droplets rapidly spread to form a thin layer with a wetting angle of less than 10 degrees. Such materials have potential applications in biosensors [3], as well as antibacterial, self-cleaning, and anti-fogging coatings [4]. Superhydrophobic surfaces are characterized by a static water contact angle exceeding 150° and a roll-off angle below 10°. A unique feature of the superhydrophobic state is the formation of air pockets in the micro/nanovoids of the multimodal surface texture, which reduces the contact area between the solid and the liquid. Thus, superhydrophobic surfaces have many practical applications in areas such as self-cleaning, corrosion protection, water–oil emulsion separation, deicing, and biofouling prevention [3,4,5,6,7,8,9]. However, widespread use is hindered by the high cost, low durability, and multistage production process of superhydrophobic materials [10].

Superhydrophobicity is achieved by combining two key factors: low surface energy at the interface and a multimodal surface topology. The influence of surface roughness on wettability is explained by two primary theories: the Wenzel model [11,12], which describes a homogeneous wetting state and considers the case when the liquid displaces air from the micro/nanoroughness of the surface, and the Cassie–Baxter model of heterogeneous wetting, which considers the situation when air remains in the micro/nanoroughness under the liquid drop [13]. Low surface energy reduces the work of adhesion and, consequently, increases the water-repellent properties of the coating [14].

To lower the surface free energy (SFE), hydrophobizing agents are used, which are low- or high-molecular-weight compounds containing long alkyl or perfluoroalkyl groups [15]. Perfluorinated compounds are preferred as modifiers owing to their low surface energy [16,17]. However, they are expensive and environmentally unfriendly. Additionally, one of the challenges in using perfluorinated compounds is their poor adhesion to substrate surfaces [18,19,20]. The use of polymer modifiers in superhydrophobic coating production is more advantageous owing to their adaptable chemical structure and composition, as well as their ability to form chemical bonds with the substrate’s functional groups, enabling covalent attachment. This considerably enhances the durability of the hydrophobic properties.

Classical methods for creating surface roughness can be categorized into physical techniques (such as plasma and laser etching, lithography, centrifugation, vapor deposition, and spraying) [21,22,23] and chemical methods (such as sol–gel process, electrochemical deposition, layered self-assembly, and acid–base etching) [24,25]. Each method offers its own advantages but also has limitations related to the complexity of the production process, the need for specialized equipment and reagents, and adherence to environmental standards.

Applying superhydrophobic coatings to products with large surfaces and complex geometries remains a considerable technological challenge. Adapting and scaling the abovementioned methods are not always feasible or economically viable. The aerosol deposition of nanoparticle suspensions or polymer binder dispersions, followed by composite coating formation, is a more versatile and efficient approach. The roughness created by this method results from the aggregation of filler nanoparticles during solvent evaporation and their interaction with the polymer binder. Particles of silicon, zinc, titanium oxides, clay minerals, calcium carbonate, and graphene are the most widely spread as fillers for creation of micro- and nanotopology [26,27,28,29,30,31,32]. However, the use of flammable and explosive organic solvents, along with their evaporative loss during curing, is a considerable drawback. In line with the trends in “green chemistry”, it is preferable to use more environmentally friendly, water-based emulsion formulations [33,34].

For example, we [35] produced superhydrophobic two-layer coatings through aerosol application of a self-curing adhesive composition made from commercial epoxy resin and silicon oxide particles functionalized with perfluorooctyltriethoxysilane. However, this method does not provide control over the coating thickness. Additionally, the adhesive base binds the nanoparticles in the inner layer, while the outer layer exhibits weak adhesion. As an alternative, prehydrophobized filler particles are introduced into adhesive compositions. In several studies [36,37,38,39], the production of superhydrophobic composite coatings using silanes, fluoropolyurethanes, polyamides, epoxy resins, and premodified silicon oxide particles as polymer binders has been demonstrated. However, a common drawback of these methods is the poor adhesion of polymer hydrophobic agents to the filler particles.

We have previously worked on the creation of superhydrophobic surfaces [40]. Fluoro- and alkyl methacrylate (AlMA) copolymers with glycidyl methacrylate (GMA) were successfully grafted onto the surface of acid–base-etched aluminum. The resulting coatings exhibited wetting angles ranging from 158° to 168° and roll-off angles of less than 5°. These copolymers contain epoxy groups that form covalent bonds with the substrate and filler surfaces and (fluoro)alkyl groups that considerably reduce SFE. Polymer coatings based on AlMA and GMA copolymers on smooth substrates achieved an SFE ranging from 35.4 to 19.7 mN/m, comparable to the SFE of 25–13 mN/m for coatings based on fluoroalkyl methacrylate copolymers with low fluorine content, ranging from 3 to 7 fluorine atoms per unit [41]. These properties make AlMA and GMA copolymers effective as polymer hydrophobic agents and film formers. It is worth developing a composite formulation using AlMA and GMA copolymers combined with inorganic fillers to create multilevel roughness suitable for single-stage aerosol application, eliminating the need for preliminary adhesive layer attachment or texturing. In this study, the silica nanoparticles were chosen as a filler due to the commercial availability of various grades with standardized size particles, high mechanical strength, thermal and chemical stability, and ease of surface modification. Additionally, aligning with “green chemistry” trends, it is crucial to minimize the release of organic solvents into the atmosphere and focus on water-based emulsion compositions. The poly(hexyl methacrylate-co-glycidyl methacrylate) [poly(HMA-co-GMA)] copolymer has been selected as the primary modifier owing to its environmental friendliness and cost-effectiveness compared to perfluorinated compounds. In this study, a one-step method of aerosol application of organo-water emulsions of poly-(HMA-co-GMA) copolymer with addition of silica nanoparticles was proposed for the formation of coatings as an “all-in-one” type without additional intermediate stages of adhesive layer fixation or substrate surface texturing, as well as preliminary surface modification of filler nanoparticles. Thus, the aim of this work is to investigate the coating formation using organic–aqueous compositions based on copolymers of AlMAs and GMA and to examine how varying the concentrations of filler and polymer binder affects the resulting roughness and hydrophobic properties of the coatings.

## 2. Materials and Methods

### 2.1. Materials

In this study, we used rectangular glass slides (25 mm × 75 mm, Micromed, St. Petersburg, Russia), analytical-grade methanol and acetone (Vekton, St. Petersburg, Russia), and a commercial solvent 646 mixture containing 10% butyl acetate, 8% ethyl cellulose, 7% acetone, 15% butyl alcohol, 10% ethyl alcohol, and 50% toluene. We also used distilled water, glycidyl methacrylate (GMA, 97%, Aldrich, St. Louis, MO, USA), hexyl methacrylate (HMA, 98%, Aldrich), azobisisobutyronitrile (AIBN, 98%, Aldrich), and silicon dioxide nanoparticles (commercial brand Aerosil A-175 with particle size 10–40 nm and specific surface area 175 ± 25 m^2^/g).

### 2.2. Synthesis of Copolymers

The poly(HMA-co-GMA) copolymer was synthesized via a free radical mechanism, following a method similar to that described previously [42]. The reaction was performed in a commercial solvent 646 at 70 °C for 24 h, with a total monomer concentration of 1 mol/L. AIBN was used as the initiator. The resulting polymer was precipitated in cold methanol and then dried under a vacuum for 24 h.

### 2.3. Preparation of Emulsion Formulations

Solutions of poly(HMA-co-GMA) copolymer were prepared in a commercial solvent 646 with concentrations of 5–20 weight percent (wt.%). According to the formulations in Table 1, filler was added to the solutions in the same mass ratio of filler to polymer binder. The suspension was stirred with a magnetic stirrer for 10 min at room temperature. Then, an equal volume of water was added, and the mixture was ultrasonically dispersed for 2 min with a Bandelin Sonopuls HD 2070 unit (Bandelin, Germany) (operating power 70 W, frequency 20 kHz, microprobe vibration amplitude 120 µm). All compositions formed an oil-in-water Pickering emulsion after dispersion.

### 2.4. Formation of Coatings

The coatings were applied following the procedure depicted in Figure 1. The schematic diagram hypothesizes that the copolymer forms microparticles (see the AFM and SEM results) and the silica nanoparticles are deposited and decorate the polymer microparticles with nanometer particles (see the AFM results). First, the glass slides were degreased with acetone and cleaned in an ultrasonic bath for 20 min. Then, the compositions were applied using an airbrush to achieve a uniform layer without gaps or streaks. The airbrush was held 200 mm from the surface and operated at a compressor pressure of 4 kgf/cm^2^. The spray was directed perpendicularly to the surface, with the spray gun moved at a consistent speed of up to 1 m/s, applying the material in a cross-hatch pattern. A total of 2 mL of the prepared emulsion was sprayed onto each glass slide. Next, the samples were dried at room temperature for 15 min before being placed in a thermal cabinet. The temperature was then gradually increased from 20 °C to 140 °C at a rate of 2 °C/min and maintained at the maximum temperature for 1.5 h. The interaction of oxirane ring with hydroxyl groups of substrates is described in the work [43], where successful modification of model inorganic surfaces (dispersed SiO_2_ particles and planar substrates Si/SiO_2_, Ge/Ge/GeO_2_, Al_2_O_3_, and TiO_2_) with poly(butadieneepoxide) has been reported. It should be noted that due to the high viscosity of the emulsion it was impossible to form the coatings by the aerosol method from compositions no. 18 and 24. All other coatings formed a homogeneous surface free of visible defects, adhered well to the substrate, and did not crumble. Evaluating the mechanical strength and abrasive wear resistance of these coatings is a labor-intensive task and will be addressed in a separate study.

### 2.5. Methods

The wetting and roll-off angles of the composite coatings were measured using an OCA 25 EC instrument (DataPhysics, Filderstadt, Germany) equipped with a tilting base unit TBU100. Deionized water droplets were applied to the substrate surface, and the contact angle of the stationary droplet was calculated using the Young–Laplace method. Statistical analysis was performed on the series of three samples of 10–15 measurements on each and the arithmetic mean of the contact angle was calculated.

To measure the roll-off angle, a 10–12 µL droplet of deionized water was placed on the sample surface. The angle of the surface was then gradually increased at a constant rate of 0.37°/s until the droplet began to roll. Statistical analysis was performed on the series of three samples of 10–15 measurements on each and the arithmetic mean of the roll-off angle was calculated.

Dynamic studies of droplet behavior on the surface of modified samples over extended time periods were performed in a chamber saturated with water vapor. Under these high-humidity conditions and without exposure to the external environment, the evaporation rate of the droplet on the modified surface was substantially reduced. This setup allowed for the observation of changes in the contact angle of a stationary droplet over long periods. The contact angle measurements were performed following the abovementioned procedure.

The coating surface morphology was evaluated using scanning electron microscopy (SEM) (Versa 3D, FEI, Morristown, NJ, USA) in low vacuum mode, with water vapor pressure in the chamber in the range of 10–80 Pa, an accelerating voltage of 10–20 kV, and beam current from 13 pA to 4 nA. The elemental composition of the surface was determined using the Oxford 51N1286 AZtecLive Expert energy-dispersive X-ray spectroscopy (EDS) instrument with an Ultim Max 65 detector (Oxford Instruments, Abingdon, UK).

Surface topology was evaluated using atomic force microscopy (AFM) (Solver Pro, NT-MTD, Moscow, Russia), with measurements performed in tapping mode. The surface of each sample was scanned at three different locations, and the AFM images were analyzed using WSxM software (v 5.0 Develop 10.2) [44].

## 3. Results

The hydrophobic properties of substrates are determined by their chemical composition and are further enhanced by surface topology. Incorporating an inorganic filler into the polymer matrix is essential for creating a hierarchical structure in the surface layer of the coating as the filler particles deposit during solvent evaporation. However, it is challenging to choose the optimal composition for achieving a uniform and stable composite coating. This involves finding the right balance between the polymer binder and the filler. Insufficient inorganic particles in the coating will result in inadequate roughness, preventing the achievement of a superhydrophobic state. Conversely, an excess of filler particles combined with a shortage of polymer binder can lead to reduced coating strength, poor adhesion to both the polymer matrix and the substrate, and ultimately, cracking [45]. Thus, developing composite water-repellent and self-cleaning coatings is a complex interdisciplinary task that involves investigating the formation of multimodal roughness and ensuring the stability of the superhydrophobic state.

We propose that the development of hydrophobic composite coatings should focus on using a polymer binder that serves not only as a polymer binder but also as an effective means of reducing SFE. Additionally, in line with current trends in “green chemistry”, which emphasize reducing perfluorinated compounds [46], we propose using functional copolymers based on AlMA. We have previously shown [41] that coatings based on these copolymers exhibit sufficiently low SFE values and can serve as an alternative to fluorinated modifiers. To form composite coatings from aqueous dispersion media, we chose a poly(HMA-co-GMA) copolymer with a short hydrocarbon substituent. The synthesis and analysis of similar functional polymers has been previously described in our works [47,48]. Elemental analysis revealed that the molar ratio of comonomers in the synthesized copolymer was [HMA]:[GMA] = 2.86:1. Gel permeation chromatography indicated that the copolymer had a low molecular weight (M_n_ = 47 × 10^3^, M_w_ = 80.5 × 10^3^) and a relatively narrow molecular weight distribution (M_w_/M_n_ = 1.7).

In the search to identify the optimal formulation for a stable superhydrophobic composite coating, the concentration of the polymer binder varied from 5 to 20 wt.%, and the mass ratio of filler to polymer was adjusted in the range of 0–2 (Table 1). The surface of the coatings was analyzed using EDS, which enabled the examination of the chemical composition of the surface layer up to a depth of 1 µm. This analysis was performed using coatings with a binder content of 10 wt.% because the filler/polymer binder mass fraction ratios (W_f_/W_p_) remained consistent across all concentrations. The results (Table 2) indicate that as the filler content in the coatings increases, the carbon content gradually decreases while the levels of silicon and oxygen increase. The presence of Na and Ca signals in the samples can be attributed to the formation of defective, uncoated areas on the substrate or to the creation of a thin coating where the depth of electron penetration by the EDS attachment exceeds the thickness of the polymer coating at those points.

Evaluating all compositions comprehensively—by examining changes in morphological features and the stability of the lyophilic properties that influence the final coating—is a complex task. Therefore, initial quality assessments were performed by studying the lyophilic properties at the phase interface. Mineral glass (commercial glass slides), which exhibits hydrophilic properties with wetting angles of 29° ± 2° and an SFE of 60.99 ± 0.5 mN/m, was used as a model substrate with a smooth surface. Figure 2 shows that polymer coatings without added particles achieve surface hydrophobization, increasing contact angles to 91.5 ± 1° and resulting in an SFE of 24.71 ± 0.9 mN/m on the glass surface. For composite coatings, measuring SFE is less accurate owing to the contribution of surface roughness. It is worth noting that increasing the concentration of the modifying copolymer poly(HMA-co-GMA) in the composition does not change the initial wetting properties owing to the formation of smooth coatings. However, changing the morphology of the coatings by introducing an inorganic filler enhances their hydrophobic properties, enabling the achievement of a superhydrophobic state with contact angles up to 159° (Figure 2). Note that compositions with W_f_/W_p_ ratio = 2 at copolymer contents of 15 and 20 wt.% turn out to be very viscous, and it was impossible to form coatings by the aerosol method. Additionally, for compositions with a polymer binder content in the range of 5–15 wt.%, the initial wetting angles remain consistent across the entire range of W_f_/W_p_ ratios (Appendix A). However, polymer composite coatings created from compositions with up to 20 wt.% copolymer content in the organic phase exhibit the smallest contact angles. This is likely attributed to the formation of a highly viscous suspension when filler particles are added, complicating the dispersion of the polymer hydrophobic agent and resulting in uneven coatings with defective areas. Owing to the low contact angle values and the lack of superhydrophobicity in coatings with 20 wt.% copolymer content, they were not investigated further.

It should be noted that the obtained coatings based on organo-water emulsions of poly-(HMA-co-GMA) copolymer with the addition of silica nanoparticles are not inferior in the value of the contact angle to similar coatings containing filler nanoparticles, as summarized in Table 3 (SiO_2_, CaCO_3_, ZnO, TiO_2_) on various substrates (Table 3).

Along with the contact angle, superhydrophobic coatings are characterized by roll-off angles, which are the tilt angles of the substrate relative to the horizontal plane at which a droplet begins to roll off the surface. High contact angles on superhydrophobic surfaces can sometimes be associated with high roll-off angles, indicating either defects in the coating and metastability in the wetting state or inherent topological features of the surface. In our previous study [41], we observed that the roll-off angle decreased with increasing angular velocity of the surface tilt. To minimize the impact of tilt pulse contribution, the studies were performed at the lowest possible angular velocity of 0.37°/s.

From the relationship between the roll-off angle (Figure 3, Appendix A) and the W_f_/W_p_ ratios, it is evident that the coatings exhibit roll-off angles ranging from 6.1° to 21.1°. Notably, for composite coatings with poly(HMA-co-GMA) copolymer concentrations of 10 and 15 wt.%, similar extreme trends are observed, with minimum roll-off angles occurring at W_f_/W_p_ = 0.8–1.2. However, for coatings with 5 wt.% binder, this range shifts to W_f_/W_p_ = 1.2–2, where the smallest roll-off angles are achieved. This shift is attributed to the reduced coating thickness and the higher filler mass required to create a multilevel microtexture. Consequently, based on the initial wetting and roll-off angle results, composite compositions with a 5 wt.% polymer binder were selected for a more detailed study of the morphological features and stability of the superhydrophobic state.

Achieving high contact angles and low roll-off angles alone does not guarantee the stability of superhydrophobic properties. The key indicator of stability is the retention of the superhydrophobic state during prolonged contact between a droplet of wetting liquid and the coating in a water-saturated atmosphere. A decrease in contact angles may indicate partial droplet evaporation or the presence of oxygen-containing groups on the coating surface, which could facilitate the adsorption of water molecules and the formation of hydrogen bonds. It is important to note that in this study, a polymer binder with a well-defined SFE was used to create the coatings. The primary factor contributing to the stability of the superhydrophobic state is the multimodal structure developed through the incorporation of filler particles.

Morphological studies were performed using SEM to investigate how filler and copolymer concentrations affect the roughness of the coatings. Figure 4 shows the SEM images of slides coated with compositions no. 1–6, as listed in Table 1. All coatings containing filler exhibit a regular hierarchical structure comprising micro- and nanosized agglomerates, which include a combination of micro- and nanoprotrusions and depressions (additional images are shown in Appendix A). As the amount of filler increases, larger particle agglomerates and the formation of bridges between the main peaks become evident in all coatings.

SEM image of the coating surface (composition 1) without filler (Figure 4A) shows small texture formed owing to solvent evaporation and destabilization of the aqueous–organic mixture. A closer examination of Figure 4 reveals that coatings based on 5 wt.% poly(HMA-co-GMA) binder exhibit defective zones (highlighted in red). For sample 2, with W_f_/W_p_ = 0.4 (Figure 4B), uncoated areas of the substrate are apparent. These uncoated regions may cause considerable changes in the wetting angle when assessing the stability of the superhydrophobic properties of the coatings. As the concentration of the Aerosil additive increased to W_f_/W_p_ = 0.8, similar uncoated areas were observed in sample 3 (Figure 4C), though to a lesser extent. The most uniform textures, free from visible defects, were observed in coatings from compositions no. 4 and 5 (Figure 4D,E), with W_f_/W_p_ = 1.2–1.6. Notably, the best roll-off angles were achieved with this specific microroughness. However, increasing W_f_/W_p_ beyond 1.6 resulted in the formation of microcracks, as observed in sample 6 (Figure 4F). This is likely attributed to an insufficient amount of copolymer to adequately bind and cover the entire surface area of the filler.

The morphology of the transverse fracture of the sample was analyzed to evaluate the coating thickness. Figure 5 shows that the coating exhibits a uniform thickness of 50–60 µm across the entire section, with a maximum observed thickness of 70 µm. Detailed examination of the structure reveals the presence of large particles that create protrusions on the surface, which are further covered by smaller nanometer-sized formations (Figure 6B).

AFM studies were performed to characterize the morphology. The following parameters were measured to quantify the surface structure of the polymer composite materials: root mean square roughness (R_q_), average roughness height (R_z_), maximum height (R_t_), kurtosis (R_ku_), and skewness (R_sk_). R_ku_ indicates whether the peaks and depressions on the surface profile are sharp or rounded. Higher R_ku_ indicates a more pronounced distribution of peaks and depressions. R_sk_ measures the asymmetry of the height distribution relative to a normal distribution and provides insights into the number and ratio of peaks and depressions on the surface. A negative value of R_sk_ indicates the prevalence of depressions in the profile, and a positive value indicates protrusions [49].

The formation of coatings results from the aggregation of filler nanoparticles interacting with a polymer binder during solvent evaporation. AFM images, scanned at both large (40 µm × 40 µm) and small (4 µm × 4 µm) surface areas (Figure 6), reveal the surface topology. The cross-sectional profiles show that the microtexture consists of large peaks covered with much smaller protrusions. This surface structure aligns well with the SEM results. To fully understand the roughness features, it is necessary to examine the quantitative indicators for both micro- and nanoscale areas in detail.

To analyze the surface mapping data from AFM, the dependencies of R_q_, R_z_, and R_t_ (Figure 7A,B), as well as R_ku_ and R_sk_ (Figure 8), on the W_f_/W_p_ ratios are plotted. As shown in Figure 7A (40 µm × 40 µm), increasing the W_f_/W_p_ ratio from 0 to 2 results in an initial increase in R_t_, which reaches a plateau of approximately 7 µm when W_f_/W_p_ = 0.8. At this point, R_q_ and R_z_ also reach their maxima (R_q_ = 1272.6 nm and R_z_ = 3572.4 nm). With a further increase in the ratio, these parameters gradually decrease to R_q_ = 618.3 nm and R_z_ = 1733.4 nm. Based on the analysis of similar dependencies in Figure 7B for the 4 µm × 4 µm region with finer texture, it is observed that the introduction of filler across all samples results in the development of a nanotopology with parameters in the following ranges: R_q_ = 106.1–151.7 nm and R_z_ = 290.9–442.4 nm. The values for maximum roughness height stabilize in the range of R_t_ = 595.7–734.8 nm as the W_f_/W_p_ ratio increases from 0.8 to 2. Thus, it can be concluded that W_f_/W_p_ = 0.8 is optimal for achieving a well-developed multimodal roughness in the composite coating.

Figure 8 shows how R_ku_ and R_sq_ change with varying W_f_/W_p_ ratios for 40 µm × 40 and 4 µm × 4 µm mapping areas. For the 40 µm × 40 µm areas, R_sk_ reaches a maximum value of 0.72 when W_f_/W_p_ = 0.4, after which it decreases towards near-zero values. This indicates that a small amount of filler creates a surface texture predominantly composed of protrusions, while higher filler concentrations result in a more balanced distribution of both depressions and peaks on the surface. Figure 8 shows a trend of decreasing R_ku_ for 40 µm × 40 µm areas as the amount of filler increases. This trend indicates a reduction in the sharpness of the protrusions and depressions in the surface texture. Of note, the roughness on the surface of microprotrusions (for 4 µm × 4 µm areas) exhibits minimal variation in R_sq_ and R_ku_, which confirms the formation of homogeneous structures with similar topologies primarily composed of nanoprotrusions.

The observed characteristics of the microtexture align well with the experimental data and account for the decrease in roll-off angles as the sharpness of the microprotrusions decreases in the 40 µm × 40 µm mapping areas. This reduction in roll-off angles is attributed to achieving a balance between the number of protrusions and depressions in the surface structure. Thus, compositions no. 3–5, based on 5 wt.% poly(HMA-co-GMA) solutions with W_f_/W_p_ = 0.8–1.6, are optimal. These compositions exhibit regular hierarchical structures without defects, showing favorable morphological features, initial contact angles, and roll-off angles.

Stability tests of the superhydrophobic state were performed on coatings with a 5 wt.% poly(HMA-co-GMA) copolymer content. As shown in Figure 9, increasing the W_f_/W_p_ ratio enhances the stability of the superhydrophobic properties of the coatings. For samples with W_f_/W_p_ = 0.4–0.8, a decrease in contact angles is observed during the first 9 h, followed by stabilization at values between 116° ± 2° and 140° ± 2°. This stabilization indicates a transition to a highly hydrophobic state, which is mainly attributed to deficiencies in surface morphology. As the W_f_/W_p_ ratio increases to 1.2–2, the superhydrophobic properties of the coating are maintained after 24 h of contact with a water droplet, showing only a slight decrease in contact angle of approximately 5° during the first 2–3 h. This decrease is attributed to droplet evaporation and the equilibrium reached when the chamber is saturated with water vapor. At these ratios, no contact spot is observed after 24 h of continuous water droplet exposure, indicating that the coating maintains a heterogeneous wetting regime and stable water-repellent properties. Therefore, a thorough examination of the morphological features, phase interface properties, and roll-off angles has enabled the identification of the optimal coating composition that ensures stability in the superhydrophobic state. Future research will focus on a more in-depth analysis of the long-term stability of these coatings.

## 4. Conclusions

The study of composite polymer coatings, created through single-stage aerosol application of aqueous organic emulsions of the poly(HMA-co-GMA) copolymer and varying concentrations of Aerosil particles (polymer binder concentration ranging from 5 to 20 wt.% and W_f_/W_p_ ratios ranging from 0 to 2), demonstrates the effect of multimodal roughness on the coatings’ hydrophobic properties. Incorporating an inorganic filler into copolymer compositions with low SFE considerably enhances the coatings’ hydrophobicity, achieving a superhydrophobic state with contact angles up to 159°. SEM and AFM morphological studies of coatings made with a 5 wt.% polymer binder concentration revealed that an optimal W_f_/W_p_ ratio of 1.2–1.6 was required to achieve multimodal roughness with minimal defects, the smallest roll-off angles of 6.1° ± 1°, and stable superhydrophobic properties over 24 h of continuous water contact. Higher filler contents resulted in coatings with a combination of nano- and microtexture features. AFM mapping of 4 µm × 4 µm areas revealed that nanotopology was characterized by stable parameter values, with R_q_ = 106.1–151.7 nm and R_z_ = 290.9–442.4 nm. The microtexture showed a trend of decreasing R_ku_ from 3.89 to 1.75 for 40 µm × 40 µm areas as filler content increased, indicating a reduction in the sharpness of peaks and depressions. These findings highlight the potential of using AlMA copolymers with GMA in developing formulations for single-stage aerosol applications, enabling the creation of superhydrophobic coatings without the need for an adhesive layer or pretexturing.

## Figures and Tables

**Figure 1 polymers-16-03094-f001:**
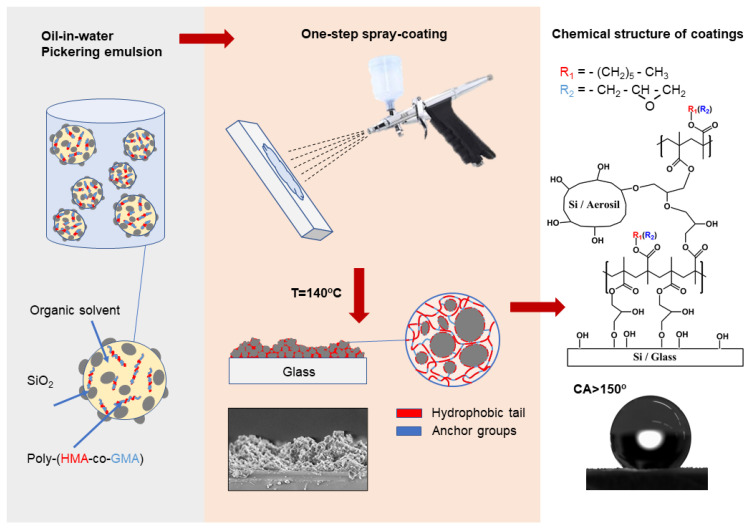
Schematic diagram of the composite coating formation.

**Figure 2 polymers-16-03094-f002:**
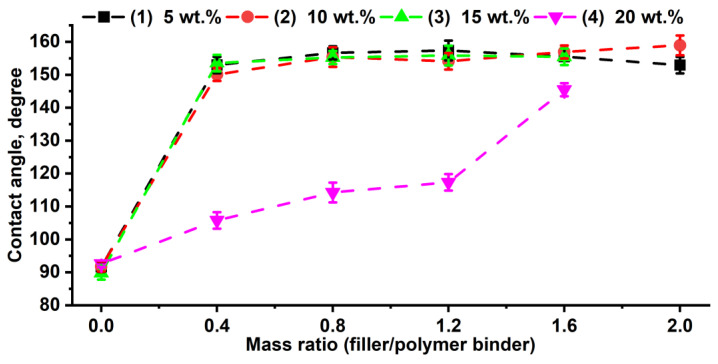
Dependence of the initial wetting angles of composite coatings based on poly(HMA-co-GMA) on the W_f_/W_p_ ratio, with varying polymer concentrations in the organic phase: (1)—5, (2)—10, (3)—15, and (4)—20 wt.%.

**Figure 3 polymers-16-03094-f003:**
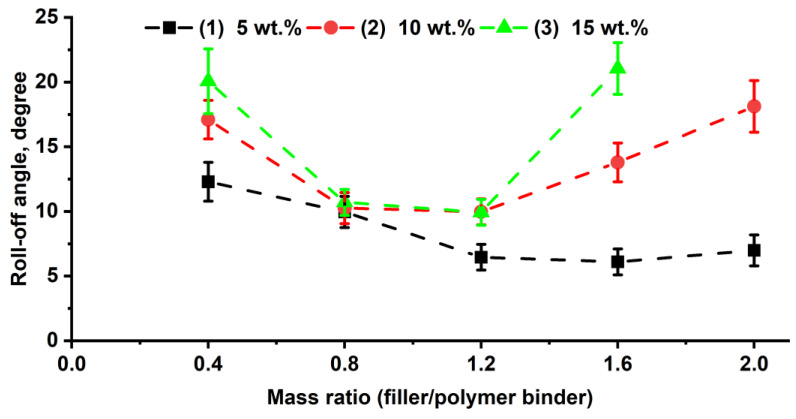
Dependence of roll-off angles on the surface of composite coatings based on poly(HMA-co-GMA) on the W_f_/W_p_ ratio, with varying polymer concentrations in the organic phase: (1)—5, (2)—10, and (3)—15 wt.%.

**Figure 4 polymers-16-03094-f004:**
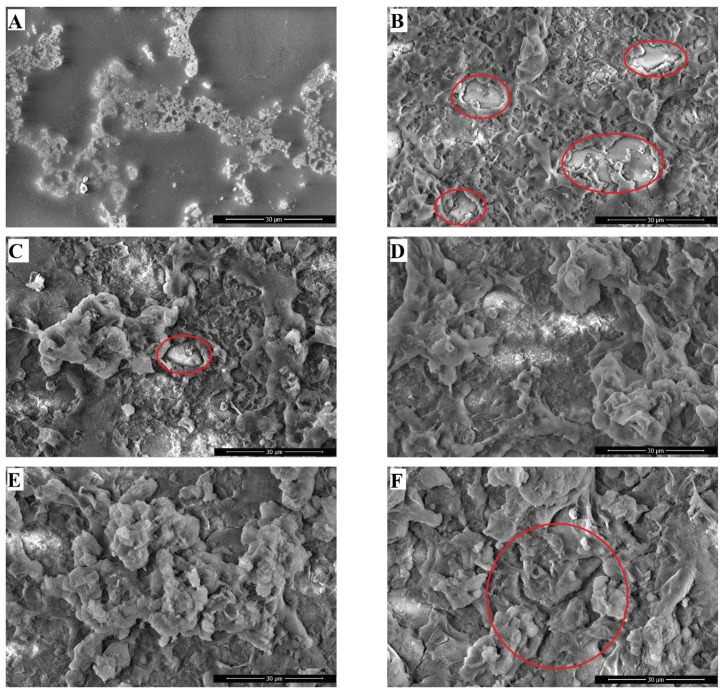
SEM images of coatings based on a 5 wt.% poly(HMA-co-GMA) solution, showing variations in the filler/polymer binder ratios: (**A**)—0; (**B**)—0.4; (**C**)—0.8; (**D**)—1.2; (**E**)—1.6; (**F**)—2.0 (4000× magnification).

**Figure 5 polymers-16-03094-f005:**
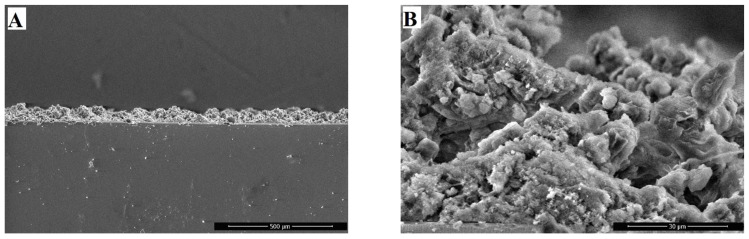
SEM image of the cross-section of a coating containing 5 wt.% poly(HMA-co-GMA) with a filler/polymer binder ratio of 1.2 (sample 4). Magnification: (**A**)—250×; (**B**)—4000×.

**Figure 6 polymers-16-03094-f006:**
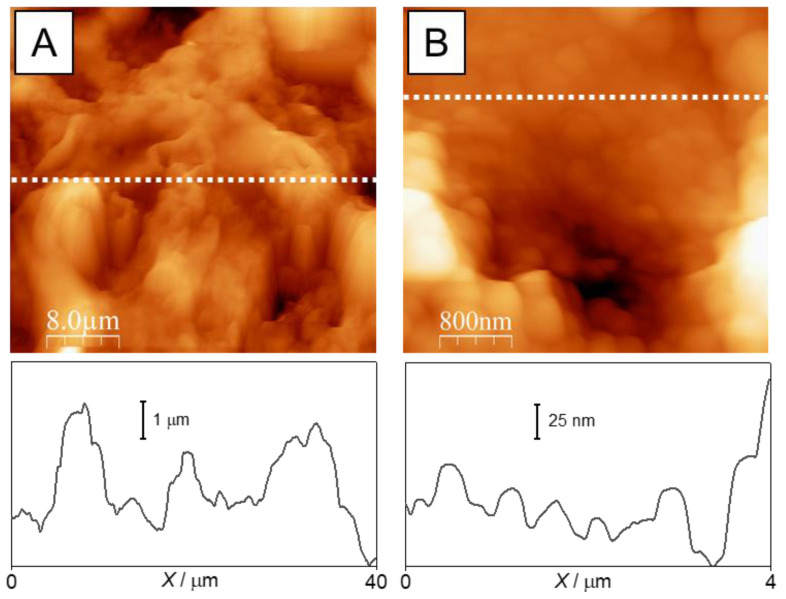
AFM images and cross-sectional profiles of the coating made from a 5 wt.% poly(HMA-co-GMA) solution with a filler/polymer binder ratio of 1.2 (sample 4). Mapping: (**A**)—40 µm × 40 µm; (**B**)—4 µm × 4 µm. The cross-sectional profiles are taken along the dotted lines indicated in the corresponding images.

**Figure 7 polymers-16-03094-f007:**
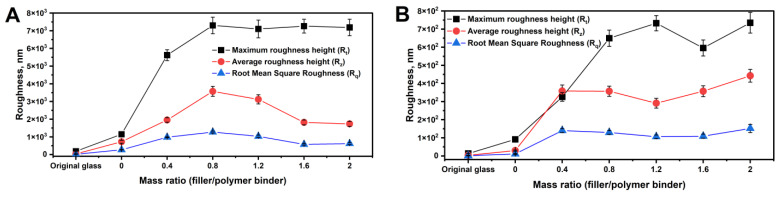
Dependence of R_t_, R_q_, and R_z_ of the composite coating (5 wt.% poly(HMA-co-GMA) solution) surface on the filler/polymer binder ratio for mapping areas: (**A**) 40 µm × 40 µm and (**B**) 4 µm × 4 µm.

**Figure 8 polymers-16-03094-f008:**
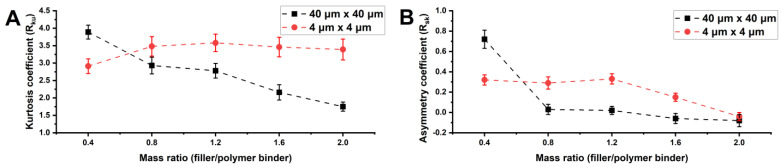
Dependence of R_ku_ (**A**) and R_sk_ (**B**) of the surface of a composite coating composed of 5 wt.% poly(HMA-co-GMA) solution on the filler/polymer binder ratio for mapping areas of 40 µm × 40 µm and 4 µm × 4 µm.

**Figure 9 polymers-16-03094-f009:**
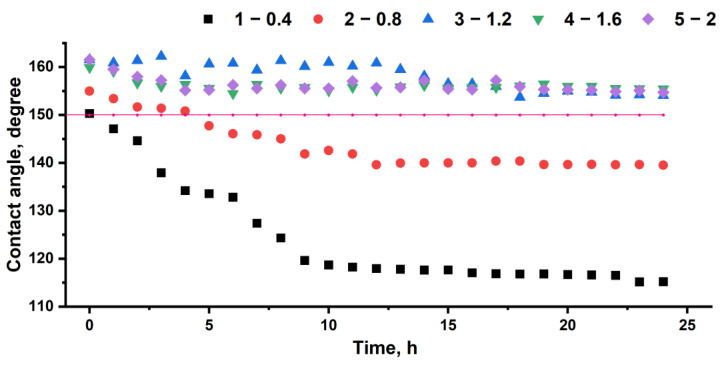
Dependence of the wetting angles on the contact time of a water droplet with the surface of polymer coatings made from 5 wt.% poly(HMA-co-GMA) solutions for various filler/polymer binder ratios: (1) 0.4; (2) 0.8; (3) 1.2; (4) 1.6; (5) 2. The pink line shows the boundary value of the wetting angle corresponding to superhydrophobicity.

**Table 1 polymers-16-03094-t001:** Formulations of the investigated compositions.

No.	Concentration of Copolymer in the Organic Phase, wt.%	Mass Content of the Components in the Mixture, %				Filler/Polymer Binder Mass Fraction Ratio, W_f_/W_p_
		Poly(HMA-co-GMA)	Aerosil	Organic Solvent	Water	
1	5%	2.4	0	44.9	52.8	0
2		2.3	0.9	44.4	52.3	0.4
3		2.3	1.9	44	51.8	0.8
4		2.3	2.8	43.6	51.3	1.2
5		2.3	3.6	43.2	50.9	1.6
6		2.3	4.5	42.8	50.4	2
7	10%	4.9	0	43.7	51.4	0
8		4.8	1.9	42.9	50.4	0.4
9		4.7	3.7	42.1	49.5	0.8
10		4.6	5.5	41.3	48.6	1.2
11		4.5	7.2	40.6	47.7	1.6
12		4.4	8.9	39.8	46.9	2
13	15%	7.5	0	42.5	50	0
14		7.3	2.9	41.3	48.5	0.4
15		7.1	5.7	40.1	47.2	0.8
16		6.9	8.3	39	45.9	1.2
17		6.7	10.7	37.9	44.6	1.6
18		6.5	13	37	43.5	2
19	20%	10.3	0	41.2	48.5	0
20		9.9	4	39.6	46.6	0.4
21		9.5	7.6	38.1	44.8	0.8
22		9.2	11	36.7	43.1	1.2
23		8.8	14.2	35.4	41.6	1.6
24		8.5	17.1	34.2	40.2	2

**Table 2 polymers-16-03094-t002:** Surface chemical composition of the coatings.

No.	Filler/Polymer Binder Mass Fraction Ratio, W_f_/W_p_	Concentration, at.%				
		C	O	Na	Si	Ca
7	0	78.42	20.23	0.33	1.03	-
8	0.4	51.27	34.37	2.26	11.23	0.87
9	0.8	54.31	32.23	0.83	12.22	0.41
10	1.2	46.54	37.45	1.41	14.09	0.52
11	1.6	42.28	40.66	0.21	16.68	0.17
12	2	41.92	39.82	0.62	17.36	0.29

**Table 3 polymers-16-03094-t003:** The comparison of hydrophobic polymer coatings containing filler nanoparticles.

№	Substrate	Polymer Binder	Filler	Modifying Composition	Filler Mass Content, wt.%	Application Method	CA, Degree	Ref.
1	Wood	Polystyrene	SiO_2_	Emulsion in acetone	1.6	Immersion in emulsion	155.6	[26]
2	Wood	1H, 1H, 2H, 2H-perfluorooctyltriethoxysilane	SiO_2_	Emulsion in ethanol	--	Spraying, immersion in emulsion, brush application	153.1 152.8 152.6	[27]
3	Glass	Polydimethylsiloxane	CaCO_3_	Solution	14.3 25 33.3 36.8	Surface spreading	120 153 160 140	[28]
4	Polyester fabric	Polydimethylsiloxane	ZnO	Solution	--	Immersion in solution	156	[29]
5	Copper	(Heptadecafluoro-1,1,1,2,2,2-tetradecyl)trimethoxysilane (FAS)/stearic acid (STA)/polydimethylsiloxane (PDMS)	TiO	Suspension	--	Immersion in suspension	162	[30]
6	Steel	Polyurethane	Aerosil R-972	Solution	8	Spraying	157	[31]
7	Steel	P(MMA-co-BA-co-DFMA-co-GMA)	SiO_2_	Solution	--	Spraying	150	[32]

## Data Availability

The data presented in this study are available on request from the corresponding authors.

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
