# Peer review of "Formation of Superhydrophobic Coatings Based on Dispersion Compositions of Hexyl Methacrylate Copolymers with Glycidyl Methacrylate and Silica Nanoparticles"

_polymers, 2024, doi:10.3390/polym16213094_

Round 1

Reviewer 1 Report

Comments and Suggestions for Authors

Abstract

Please, add 1-2 sentences about actuality of research.

The abstract is not informative, it does not reveal the full essence of the work.

Matherials and methods

It is necessary to add information how the statistical analysis of data was carried out.  

Results

The authors should describe the mentioned models in the references 11-13.

It is necessary to add error bar on the pictures.

Have the authors tried to take a different ratio of the sizes of copolymers by molecular weight? It would be interesting to see the effect with a wider molecular-weight distribution. Why were these ratio and copolymers mass chosen?

It is necessary to demonstrate on the picture 2 more detailed researchers to talk about any dependencies. The authors did not add error bar. They should be indicated everywhere. Moreover, the wider range of concentration should be used. For example, more 20% and between 15% and 20%.

On the graphs for 15 and 20% the last points of measurement are absent. What is the reason? Especially that for 20% there is a tendency for a sharp increase in wetting angles on the ratio Wf/Wp.

Picture 3 needs more details. The results for 15% and 20% are ambiguous. What explains the more intense drop in roll angles at 15 and 10% in the Wf/Wp range of 0.4-0.8?

Author Response

We appreciate the opportunity to present a revised version of the manuscript “Formation of superhydrophobic coatings based on dispersion compositions of hexyl methacrylate copolymers with glycidyl methacrylate” for publication in the Polymers. We are grateful for the time and effort Academic editor and the reviewers invested in thoroughly evaluating our work and the insightful comments and valuable improvements to our paper. We have studied comments carefully and have made corrections which we hope meet with approval. Below is a point-by-point response to the reviewers’ comments and concerns.

  1. Abstract. Please, add 1-2 sentences about actuality of research. The abstract is not informative, it does not reveal the full essence of the work.

Reply: Thank you very much for your guidance. We have revised the text of the abstract.

  1. Materials and methods. It is necessary to add information how the statistical analysis of data was carried out.

Reply: Thank you for the valuable comment. We have added the description to the text of the manuscript.

  1. Results. The authors should describe the mentioned models in the references 11-13. It is necessary to add error bar on the pictures.

Reply: Thank you for bringing this point. In the introduction we have added a description of wetting models. The visualization of errors was added at the figures.

  1. Have the authors tried to take a different ratio of the sizes of copolymers by molecular weight? It would be interesting to see the effect with a wider molecular-weight distribution. Why were these ratio and copolymers mass chosen?

Reply: Thank you for your question. In this work we have not varied the structure and molecular weight characteristics of copolymers. Our research was aimed to find the optimal concentration of filler and polymer binder, study of its influence on the roughness and final properties of the coating. At the next works we plan to use copolymers with higher molecular weight and different structures to study the influence of its contribution to the coatings properties.

  1. It is necessary to demonstrate on the picture 2 more detailed researchers to talk about any dependencies. The authors did not add error bar. They should be indicated everywhere. Moreover, the wider range of concentration should be used. For example, more 20% and between 15% and 20%.

On the graphs for 15 and 20% the last points of measurement are absent. What is the reason? Especially that for 20% there is a tendency for a sharp increase in wetting angles on the ratio Wf/Wp.

Reply: Thank you very much for your guidance. Our fault was that we forgot to add errors to the graphical dependencies (corrections were made in the figures). Compositions with the ratio Wf/Wp=2 with copolymer content of 15 and 20 wt. % turned out to be very viscous and it was impossible to form coatings by aerosol method. Expansion of the range of copolymer content concentrations over 20 wt% is impossible due to reaching the filler concentration limit.

  1. Picture 3 needs more details. The results for 15% and 20% are ambiguous. What explains the more intense drop in roll angles at 15 and 10% in the Wf/Wp range of 0.4-0.8?

Reply: Thank you for your question. For all binder concentrations the values of roll-off angles in the range of Wf/Wp 0.4-0.8 are quite close. We can assume that for compositions in the range of Wf/Wp 0.4-0.8 with binder concentrations of 10% and 15%, the formation of different microrelief structures is observed due to the influence of viscosity, which is shown in the behavior of roll-off angles.

Reviewer 2 Report

Comments and Suggestions for Authors

1.       Please add the reason for conducting the research and the novelty of the work in the abstract section.

2.       In the first paragraph in the introduction section, it is better to compare superhydrophilicity and superhydrophobicity and explain superhydrophobicity characteristics and superiority.

3.       In the introduction section, please improve the second paragraph by providing reports.

4.       In the third paragraph of the introduction section, please provide reports on the use of polymers as examples.

5.       In the fourth paragraph of the introduction section, it is better to explain the method you used.

6.       In the last paragraph of the introduction section, please give more details about the work done.

7.       The first figure (Figure 1) is confusing. Please improve the shape.

8.       Please provide optical images of the optimum sample before and after coating.

9.       In figures 2 and 3, it is confusing to specify the graphs as 1-5 wt.%. Please rewrite them as 5 wt.%.

10.   Please justify the result obtained about the waster contact angle of the samples using the EDS table.

11.   Please provide pictures of the samples with drops on them.

12.   It is better to present the SIM images at the nano scale in Figures 4 and 5. Also, please relate the results obtained from the images to superhydrophobicity results.

13.   Please justify the results obtained in superhydrophobicity using AFM results.

Comments on the Quality of English Language

 Moderate editing of English language required.

Author Response

We appreciate the opportunity to present a revised version of the manuscript “Formation of superhydrophobic coatings based on dispersion compositions of hexyl methacrylate copolymers with glycidyl methacrylate” for publication in the Polymers. We are grateful for the time and effort Academic editor and the reviewers invested in thoroughly evaluating our work and the insightful comments and valuable improvements to our paper. We have studied comments carefully and have made corrections which we hope meet with approval. Below is a point-by-point response to the reviewers’ comments and concerns.

  1. Please add the reason for conducting the research and the novelty of the work in the abstract section.

Reply: Thank you very much for your guidance. We have revised the text of the abstract.

  1. In the first paragraph in the introduction section, it is better to compare superhydrophilicity and superhydrophobicity and explain superhydrophobicity characteristics and superiority.

Reply: Thank you for your valuable comment. We have made revisions to the first paragraph of the introduction.

  1. In the introduction section, please improve the second paragraph by providing reports.

Reply: Thank you for your helpful comments. We made additions to the second paragraph of the introduction and included a description the wetting models.

  1. In the third paragraph of the introduction section, please provide reports on the use of polymers as examples.

Reply: We thank you for the important comment. We added a table comparing similar superhydrophobic coatings described in the literature on different types of substrates using nanoparticle fillers (SiO2, CaCO3, ZnO, TiO2) and the coatings obtained in this work.

  1. In the fourth paragraph of the introduction section, it is better to explain the method you used.

Reply: We are thankful for your feedback. We have made additions to the text of the fourth and last paragraph of the introduction.

  1. In the last paragraph of the introduction section, please give more details about the work done.

Reply: We greatly appreciate this question, as it highlights the need for a more precise emphasis on the outcomes of our work. At the last paragraph of the introduction the extended information about the work we have done was added.

  1. The first figure (Figure 1) is confusing. Please improve the shape.

Reply: Thank you for your valuable comment. We have made corrections to the summary process scheme shown in Figure 1.

  1. Please provide optical images of the optimum sample before and after coating.

Reply: Thank you for the comment. Images of the surface of the samples before and after modification have been added to the supplementary materials.

  1. In figures 2 and 3, it is confusing to specify the graphs as 1-5 wt.%. Please rewrite them as 5 wt.%.

Reply: Thank you for the comment. Corrections have been made to Figures 2 and 3.

  1. Please justify the result obtained about the waster contact angle of the samples using the EDS table.

Reply: Thank you for your interesting question. The main chemical elements for the investigated composite coatings are: carbon, oxygen and silicon. However, the varying of filler amount in the composition leads to an increase in detectable silicon and oxygen. We would like to note that on these samples it is not correct to draw conclusions about wetting angles on the basis of EDX data, due to the lack of indicator elements such as fluorine.

  1. Please provide pictures of the samples with drops on them.

Reply: Thank you for the comment. Photos of samples with water droplets on the surface have been added to the supplementary materials.

  1. It is better to present the SIM images at the nano scale in Figures 4 and 5. Also, please relate the results obtained from the images to superhydrophobicity results.

Reply: Thank you for the comment. In supplementary materials we have added a series of SEM images at higher magnification for samples with different content of polymer binder and filler.

  1. Please justify the results obtained in superhydrophobicity using AFM results.

Reply: Thank you for raising this important point. The structure of the surface nano-roughness determines the surface wetting mode for constant values of surface free energy. In our work in the section devoted to the study of quantitative parameters of nanoroughness, Figure 7b demonstrates how the topology parameters change in the ranges: Rq=106.1÷151.7 nm and Rz=290.9÷442.4 nm when a filler is introduced to the composition with a binder concentration of 5%. The graph shows a sharp jump in the order of Rq in comparison with the sample without filler, that well agrees with the results in Figure 2, where there is a sharp increase in the values of contact angles at the introduction of filler.

Reviewer 3 Report

Comments and Suggestions for Authors

The manuscript "Formation of superhydrophobic coatings based on dispersion compositions of hexyl methacrylate copolymers with glycidyl
methacrylate" reports the evaluation of filler concentration on the hydrophobic properties and surface roughness of composite coatings made from organo–aqueous compositions based on hexyl methacrylate (HMA) and glycidyl methacrylate (GMA) copolymers. The paper is important for the coatings field and generates a number of other experimental details that the authors mentioned to present in other paper. From this point of view the paper in not complete. However, there are a series of results which may constitute the core problem in composite films. 

The following issues must be addressed:

Abstract: Filler must be mentioned.

Line 21- the optimal mass fraction ratio must be mention

The novelty of the study must be highlighted.

Introduction: The authors highlighted the importance of superhydrophobic surfaces, but I consider necessary a Table containing the main superhydrophobic polymers/ co-polymers used in coatings.

-the role of the fillers must be emphasized

-the motivation of choosing SiO2 particles must be mentioned;

-the comparisons with other nanocompozite materials based on similar/ different polymeric matrices must be added.

Section 2.1.

The properties of the filler must be added: size, porosity, etc.

Section 2.2. The chemical reactions to prepare the copolymer must be added.

How was monitored the reaction product? FT-IR and NMR analyses are missing.

Fig. 1- How was emphasized the chemical bond between Aerosil and polymeric matrix?

Table 2. Ca and Na are not necessary to be added!

How was estimated the uniform distribution of the filler in the matrix? By which parameters could be directed to obtain uniform distributed particles in the composite?

I recommend the publication of this paper after Major revision!

Author Response

We appreciate the opportunity to present a revised version of the manuscript “Formation of superhydrophobic coatings based on dispersion compositions of hexyl methacrylate copolymers with glycidyl methacrylate” for publication in the Polymers. We are grateful for the time and effort Academic editor and the reviewers invested in thoroughly evaluating our work and the insightful comments and valuable improvements to our paper. We have studied comments carefully and have made corrections which we hope meet with approval. Below is a point-by-point response to the reviewers’ comments and concerns.

  1. Abstract: Filler must be mentioned.

Line 21- the optimal mass fraction ratio must be mention

Reply: Thank you very much for your guidance. We have revised the text of the abstract.

  1. The novelty of the study must be highlighted.

Reply: We greatly appreciate this question, as it highlights the need for a more precise emphasis on the outcomes of our work. At the last paragraph of the introduction the extended information about the work we have done was added.

  1. Introduction: The authors highlighted the importance of superhydrophobic surfaces, but I consider necessary a Table containing the main superhydrophobic polymers/ co-polymers used in coatings.

-the role of the fillers must be emphasized

-the motivation of choosing SiO2 particles must be mentioned;

-the comparisons with other nanocompozite materials based on similar/ different polymeric matrices must be added.

Reply: Thank you for pointing out this missing block of information. In the introduction, we have added additional information on the role of filler in the coatings and on the choice of filler particles. Also, the table comparing similar superhydrophobic coatings described in the literature on different types of substrates using nanoparticle fillers (SiO2, CaCO3, ZnO, TiO2) and the coatings obtained in this work was added.

  1. Section 2.1. The properties of the filler must be added: size, porosity, etc.

Reply: Thank you for comment. We have added information about Aerosil-175 in the material description section.

  1. Section 2.2. The chemical reactions to prepare the copolymer must be added.

How was monitored the reaction product? FT-IR and NMR analyses are missing.

Reply: Thank you for raising this important point. Synthesis of similar functional polymers has been previously described in our works, and references to the studies were indicated in the text of the manuscript. The chemical scheme of the synthesis was not added to the text only because we do not want to duplicate our previous works.

  1. Fig. 1- How was emphasized the chemical bond between Aerosil and polymeric matrix?

Reply: We are grateful for your question. The issue of chemical grafting of copolymers is central to our series of works. At this stage, our research group is studying the of grafting regularities (search for chemical interactions) of polymers on the surface of substrates using a series of anchor compounds with a set of physicochemical methods of analysis (XPS, solid-NMR, etc.). We plan to publish a separate article on the results of this study. To answer your question, we have added a link to a well-known article on polymer coatings by the scientific group H. Adler (Köthe M. et al. Examination of poly(butadiene epoxide)-coatings on inorganic surfaces // Colloids and Surfaces A: Physicochemical and Engineering Aspects. 1999. Vol. 154, â„– 1-2. P. 75-85.).

  1. Table 2. Ca and Na are not necessary to be added!

Reply: Thank you for your comment. You are absolutely right, Ca and Na are impurity elements, but we decided to leave it so as not to raise additional questions from other reviewers.

  1. How was estimated the uniform distribution of the filler in the matrix? By which parameters could be directed to obtain uniform distributed particles in the composite?

Reply: Thank you for your question. All compositions were intensively dispersed using ultrasound before coating. SEM images of the surface and cross-section of the samples show that the coatings are quite homogeneous. We would like to note that during the formation of coatings, the formation of agglomerates in the polymer matrix is inevitable, providing micro/nano roughness (SEM and AFM data).

Round 2

Reviewer 1 Report

Comments and Suggestions for Authors

The authors took into account most of comments and answered on all requests persuasively. I recommend the article for submition. However, there are one more question mentioned in the Notice â„– 5 below.

On the graphs for 15 and 20% the last points of measurement are absent. What is the reason? Especially that for 20% there is a tendency for a sharp increase in wetting angles on the ratio Wf/Wp.

Author Response

We thank the Reviewer for careful reading of our manuscript, positive evaluation and valuable comments and questions which certainly helped us to improve the manuscript. 

1) The authors took into account most of comments and answered on all requests persuasively. I recommend the article for submition. However, there are one more question mentioned in the Notice â„– 5 below.

On the graphs for 15 and 20% the last points of measurement are absent. What is the reason? Especially that for 20% there is a tendency for a sharp increase in wetting angles on the ratio Wf/Wp.

Reply: Thank you for the important question. The reason for this is high viscosity of compositions with Wf/Wp=2 at copolymer content of 15 and 20 wt. % and therefore it was not possible to form coatings by aerosol method. We made an addition to the text of the article (highlighted in green, L257).

Reviewer 2 Report

Comments and Suggestions for Authors

Accept in present form

Comments on the Quality of English Language

 Minor editing of English language required.

Author Response

We thank the Reviewer for careful reading of our manuscript, positive evaluation and valuable comments and questions which certainly helped us to improve the manuscript. 

Reviewer 3 Report

Comments and Suggestions for Authors

The revised manuscript was improved according to the suggestions and can be considered for publication!

Author Response

(The authors gave the same response as above.)
